# scQTLtools: An R/Bioconductor Package for Comprehensive Identification and Visualization of Single-Cell eQTLs

**DOI:** 10.3390/biology14070743

**Published:** 2025-06-23

**Authors:** Xiaofeng Wu, Xin Huang, Pinjing Chen, Jingtong Kang, Jin Yang, Zhanpeng Huang, Siwen Xu

**Affiliations:** 1School of Medical Information and Engineering, Guangdong Pharmaceutical University, Guangzhou 510006, China; 2100507148@stu.gdpu.edu.cn (X.W.);; 2Guangdong Province Precise Medicine Big Data of Traditional Chinese Medicine Engineering Technology Research Center, Guangdong Pharmaceutical University, Guangzhou 510006, China

**Keywords:** single-cell eQTL analysis, eQTL identification, R, Bioconductor, single-cell RNA-seq, cis-regulatory variants, genotype–expression association

## Abstract

Every person is different, and understanding how our genes influence health and disease is a key goal of modern science. However, traditional methods often study mixed groups of cells, which can hide important genetic effects. In this study, we developed a new computer tool that helps scientists explore how genetic differences affect gene activity in individual cells. This tool makes it easier for researchers to process and analyze complex data by providing clear steps and interactive plots. We tested our tool on the dataset from a type of blood cancer and found that some genetic changes only affect certain cell types. These findings show how important it is to look at cells one by one rather than in bulk. Our tool can help researchers discover new disease-related genes and better understand how illnesses develop in different parts of the body. In the future, this may lead to more accurate diagnoses and personalized treatments for patients.

## 1. Introduction

The study of gene regulation is fundamental to understanding cellular function and disease mechanisms [1]. Single-cell RNA sequencing (scRNA-seq) has revolutionized the field by providing insights into gene expression at an unprecedented resolution, allowing researchers to study the heterogeneity within complex tissues [2,3]. However, gene expression is not solely determined by transcriptional regulation; genetic variants also play a crucial role. Expression quantitative trait loci (eQTLs) studies have been instrumental in linking genetic variants to gene expression levels, but traditional eQTLs analyses are typically limited to bulk tissue samples, which can obscure cell-type-specific effects [4].

Recent advances in single-cell multi-omics technologies, such as joint profiling of gene expression and chromatin accessibility, now make it possible to simultaneously measure different omics within the same cell [5]. This allows for integrative analyses of gene expression and genotype at the single-cell level, avoiding the ambiguity associated with data derived from different cells and enhancing the resolution of regulatory inference.

The advent of single-cell technologies has also opened new avenues for eQTLs analysis, enabling the investigation of genetic effects at the resolution of individual cells [6]. Single-cell eQTLs (sc-eQTLs) analysis provides a unique opportunity to uncover genetic influences on gene expression that are specific to particular cell types or states, providing a more detailed understanding of regulatory mechanisms. For instance, van der Wijst et al. identified 379 unique eQTLs from peripheral blood mononuclear cells, of which 48 eQTLs were detected exclusively in specific cell types and not in any eQTLs from bulk RNA-seq data [7]. Similarly, Neavin et al. performed sc-eQTLs analysis on fibroblasts and induced pluripotent stem cells (iPSCs), demonstrating that most of the detected eQTLs in fibroblasts were specific to individual cell subtypes [8]. Notably, less than half of the eQTLs identified in their study were reported by GTEx, a canonical public resource for eQTLs. These findings highlight the significant cell-type-specific gene regulatory insights that sc-eQTLs analysis can provide, which are often missed in bulk QTLs mapping. Thus, sc-eQTLs analysis can substantially improve eQTLs detection compared with bulk sequencing data.

Despite these advancements, sc-eQTLs analysis presents several challenges, including the sparsity and technical noise inherent in scRNA-seq data, the high cost of single-cell sequencing, and the need for sophisticated statistical models to detect eQTLs with sufficient power [9]. Several tools have been developed to address these challenges. For example, Matrix eQTL, a highly efficient method for bulk eQTLs analysis developed by Shabalin, lacks the sensitivity required for single-cell data [10]. SCeQTL, which employs a zero-inflated negative binomial regression method optimized for single-cell data, addresses some of these limitations but lacks critical preprocessing functions, such as data filtering and normalization [11]. Another recent tool, eQTLsingle, supports sc-eQTLs detection but only supports binary genotype grouping, which can lead to differences in statistical power and visual interpretation compared with the more commonly used three-class genotype approach (homozygous reference allele, heterozygous, and homozygous alternative allele) [12].

To fill these gaps, we present scQTLtools, an R package (v1.1.7) designed for streamlined sc-eQTLs analysis and visualization. Our package integrates state-of-the-art methods for eQTLs identification with comprehensive data preprocessing and multiple visualization options, providing researchers with a powerful tool for exploring sc-eQTLs within specific cellular contexts. By facilitating the detailed exploration of genetic regulation at single-cell resolution, scQTLtools aims to advance our understanding of gene regulation in health and disease.

## 2. Materials and Methods

### 2.1. Overview of scQTLtools

scQTLtools is implemented as an R package designed for streamlined sc-eQTLs analysis and visualization. The package is developed to provide researchers with a comprehensive, user-friendly toolkit for identifying and interpreting sc-eQTLs within complex single-cell datasets. The core functionality of scQTLtools includes data preparation, normalization, quality control, statistical modeling, and result visualization (Figure 1). The package is structured to be flexible and modular, allowing users to customize their analyses based on their specific research goals. scQTLtools is freely available from Bioconductor (https://bioconductor.org/packages/scQTLtools/ (accessed on 30 May 2025)). The source code is available and maintained on GitHub (https://github.com/XFWuCN/scQTLtools (accessed on 30 May 2025)).

### 2.2. Data Preparation and Input Functions

To initiate the analysis pipeline, scQTLtools provides the createQTLObject() function, which constructs an S4 object that serves as the standardized container for both input data and cell group metadata (Figure 1a). This object is required for all downstream analysis functions. The function integrates a single-cell gene expression matrix and a corresponding SNP genotype matrix, along with optional sample annotations.

In practice, the gene expression matrices output by CellRanger from 10x Genomics single-cell RNA-seq data can be directly imported into R as Seurat or SingleCellExperiment objects and used as input for scQTLtools without additional reformatting.

The input genotype matrix should be organized with rows representing SNPs and columns representing cells and should follow the following encoding scheme: 1 indicates a homozygous reference genotype (e.g., AA), 2 indicates a homozygous alternative genotype (e.g., GG), 3 indicates a heterozygous genotype (e.g., AG or GA), and 0 indicates missing values. This encoding scheme is commonly adopted in eQTL studies based on reference genomes, where alleles are defined relative to the reference sequence. Considering the high sparsity of single-cell data, scQTLtools also supports a simplified encoding mode via the *biClassify* parameter, where 1 indicates the absence of variant (reference genotype), 2 indicates the presence of variant (non-reference genotype), and 0 denotes missing values. This alternative encoding is compatible with binary and additive genetic models and is particularly useful for linear-regression-based methods applied to sparse datasets. We recommend using tools such as VarTrix (10x Genomics, v1.1.22) [13] for generating genotype matrices. These tools ensure that the genotype matrix matches the cell barcodes in the expression matrix, facilitating streamlined data integration.

### 2.3. Gene Expression Normalization

To correct for technical variability and ensure the comparability of gene expression across cells, scQTLtools implements the normalizeGene() function for robust normalization of single-cell gene expression data (Figure 1b). This step is essential for reducing the effects of sequencing depth, capture efficiency, and other confounding technical biases that are common in scRNA-seq datasets.

The normalizeGene() function takes an input object of class eQTLObject and returns an updated eQTLObject in which the gene expression matrix has been normalized according to the specified method. scQTLtools supports five commonly used normalization methods via the normalizeGene() function: log-normalization (*logNormalize*), counts per million (*CPM*), transcripts per million (*TPM*) [14], DESeq2-based normalization (*DESeq*) [15], and limma-based normalization (*limma*) [16]. Each normalization method is internally implemented to handle sparse gene expression matrices and can be used directly on data stored in Seurat or SingleCellExperiment format within the eQTLObject. The output is a normalized matrix of gene expression values that can be accessed and used in downstream eQTL mapping functions.

### 2.4. Quality Control and Filtering

Single-cell RNA-seq and genotype matrices are often characterized by sparsity and technical noise, which may compromise the reliability of eQTL analysis if left unfiltered. To address this, scQTLtools provides the filterGeneSNP() function to perform quality control (QC) by jointly filtering the gene expression and SNP genotype matrices based on user-defined thresholds (Figure 1b).

The function operates on an input eQTLObject and applies two levels of filtering:

(1) Gene-level filtering: Genes are retained if their expression exceeds the threshold defined by *expressionMin* in at least the proportion of cells specified by the *expressionNumOfCellsPercent* parameter.

(2) SNP-level filtering: SNPs are retained if each genotype group (e.g., homozygous reference, heterozygous, homozygous alternative) is represented in at least the percentage of cells defined by the *snpNumOfCellsPercent* parameter. This helps eliminate genotype sites that lack representation across diverse cell populations and would otherwise reduce the power of association testing. By default, SNPs are filtered to retain those where each genotype class occurs in at least 10% of the cells.

The filtered results are stored back in an updated eQTLObject, ensuring seamless integration with downstream modeling and visualization functions.

### 2.5. sc-eQTLs Mapping and Statistical Models

To identify eQTLs at single-cell resolution, scQTLtools implements the callQTL() function, which provides an integrated framework for sc-eQTLs detection using only scRNA-seq data and user-supplied genotypes (Figure 1c). This function combines genomic proximity filtering, flexible statistical modeling, and multiple testing correction to uncover significant SNP–gene associations.

The function operates on an input object of class eQTLObject and optionally allows users to restrict the analysis to a subset of genes via the *gene_ids* argument. To define candidate cis-regulatory relationships, callQTL() identifies SNPs located within a specified upstream or downstream window (in base pairs) of each gene, as defined by the *upstream* and *downstream* arguments. Gene and SNP coordinates can be retrieved in real time using the Ensembl BioMart system, specified through the *gene_mart* and *snp_mart* arguments. If these are not explicitly provided, the function will automatically connect to default Ensembl datasets.

For each SNP–gene pair, the association between genotype and gene expression is modeled using the method specified in the *useModel* argument. scQTLtools supports three commonly used models: linear regression (*linear*) [10,17], Poisson regression (*poisson*) [17], and zero-inflated negative binomial regression (*zinb*). The ZINB model, used by default, is particularly effective in addressing the excess zero inflation characteristic of sparse UMI-based scRNA-seq data [18,19]. All these models are commonly used for eQTL mapping in single-cell and bulk transcriptomic studies. Model selection can be guided by gene-level expression characteristics, where the linear model is suitable for normalized continuous data with low sparsity, Poisson regression fits count data with limited dispersion, and the ZINB model is recommended for highly sparse or zero-inflated expression profiles.

To control for multiple hypothesis testing, callQTL() applies *p*-value adjustment methods via the *pAdjustMethod* argument. Supported options include *bonferroni*, *holm*, *hochberg*, *hommel*, and *BH* (Benjamini–Hochberg). Significant eQTLs are defined as those with adjusted *p*-values below *pAdjustThreshold* (default: 0.05) and passing the effect-size threshold set by *logfcThreshold* (default: 0.1).

The output is a structured result object that contains SNP–gene pairs along with their estimated effect sizes, raw and adjusted *p*-values, and optional genomic annotations. These results can be used directly for visualization or downstream biological interpretation.

### 2.6. Visualization of eQTL Results

To facilitate intuitive interpretation of SNP–gene associations, scQTLtools provides the visualizeQTL() function for flexible visualization of eQTL results across single-cell groups. This function supports multiple plot types and offers customizable options for group selection and outlier handling (Figure 1d).

The visualizeQTL() function operates on an eQTLObject and takes as input a target SNP ID (*SNPid*) and gene ID (*Geneid*). Users can optionally specify one or more groups of interest via the *groupName* argument to visualize cell-type-specific or condition-specific associations. Four plotting options are supported through the plottype argument: *QTLplot*, *violin*, *boxplot*, and *histplot*, each designed to highlight different aspects of the genotype–expression relationship.

Additionally, outlier removal can be enabled via the *removeoutlier* argument, which identifies cells with expression values exceeding the median plus four times the median absolute deviation (MAD). These cells are flagged as outliers and excluded from the plot to reduce the influence of extreme values in sparse single-cell datasets.

Internally, visualizeQTL() leverages the ggplot2 (v3.5.1) [20] package for generating high-quality publication-ready plots and integrates with patchwork (v1.2.0) [21] for composing multi-panel layouts when multiple groups are selected. This flexible visualization framework enables researchers to explore genotype-dependent expression patterns and visually validate candidate eQTLs in specific cellular contexts.

## 3. Results

### 3.1. Comparison with Existing eQTL Analysis Tools

To evaluate the features and advantages of scQTLtools, we compared it with several existing tools for eQTL analysis, including eQTLsingle, SCeQTL, Matrix eQTL, and iBMQ [22] (Figure 2). All tools were assessed based on key functionalities, including input compatibility, built-in reference genome annotations, genotype encoding flexibility, preprocessing support, model diversity, and visualization capabilities. In the comparison table, the first column lists detailed sub-features under broader categories, and each checkmark indicates the presence of a given feature, while “NA” denotes its absence.

Among these tools, scQTLtools offers the most comprehensive feature set. Notably, it supports both Seurat and SingleCellExperiment object formats as input, which are standard data structures for single-cell RNA-seq analysis in R. While some tools are limited to binary genotype matrices, scQTLtools accommodates both binary and three-class genotype encodings (i.e., homozygous reference, heterozygous, and homozygous alternative), thereby increasing its compatibility with diverse genetic datasets.

In terms of preprocessing, scQTLtools provides integrated quality control functions for filtering low-quality SNPs and genes, along with multiple normalization methods to account for technical biases in gene expression data. Users can also customize the genomic window for defining SNP–gene pairs, enabling flexible cis-eQTL discovery.

For statistical modeling, scQTLtools offers three fitting options—linear regression, Poisson regression, and ZINB—which accommodate various data distributions and analysis needs. The inclusion of model selection provides users with flexibility to tailor the analysis based on the sparsity and characteristics of their dataset.

Furthermore, scQTLtools incorporates a suite of visualization tools for exploring eQTL results at single-cell resolution. These include scatter plots, boxplots, violin plots, and histograms, all of which support stratification by cell type or condition. This is particularly advantageous for dissecting cell-type-specific regulatory effects in heterogeneous tissues.

Compared with SCeQTL and eQTLsingle, which are explicitly developed for single-cell eQTL analysis, scQTLtools provides several practical advantages that enhance usability and analytical robustness. First, it supports a broader range of genotype encoding strategies (0/1 and 0/1/2), improving compatibility with standard genotype data formats. Second, it integrates a comprehensive preprocessing pipeline—including gene and SNP filtering, normalization, and cell-type grouping—allowing users to perform key preparatory steps without relying on external R packages. Finally, its diverse set of built-in visualization modules facilitates the intuitive interpretation of eQTL results across heterogeneous cell populations.

To evaluate the computational performance of scQTLtools, we benchmarked runtime and memory usage for three statistical models (linear, Poisson, and ZINB) using the AML dataset. The results are summarized in Appendix A. Furthermore, we performed a sensitivity analysis by artificially increasing sparsity through downsampling of the gene expression matrix. We observed that model performance, in terms of runtime and memory consumption, varied with both model complexity and data sparsity. These results provide practical guidance for model selection and resource planning when applying scQTLtools to datasets of varying sizes and sparsity levels.

Taken together, scQTLtools combines flexibility, compatibility, and usability in a unified R package, making it a powerful tool for sc-eQTL analysis. Its broad feature coverage and seamless integration with single-cell data workflows distinguish it from existing methods.

### 3.2. Case Study on Human Acute Myeloid Leukemia

To demonstrate the functionality and practical utility of scQTLtools, we applied it to a single-cell multi-omics dataset from human acute myeloid leukemia (AML). Specifically, we analyzed three AML samples with matched single-cell genotype and gene expression profiles. The cells span major hematopoietic lineages, including hematopoietic stem cells (HSCs), common myeloid progenitors (CMPs), granulocyte-monocyte progenitors (GMPs), and monocytes. Based on cell abundance, we selected CMPs and GMPs as the primary cell groups for eQTL analysis. For each gene, we systematically scanned the genome to identify nearby SNPs that may influence its expression through cis-regulatory mechanisms.

At each step of the analysis, scQTLtools produced an updated eQTLObject, which encapsulated all relevant components, including the raw input matrices, filtered data, intermediate results, final eQTL mappings, and associated cell group annotations. During eQTL mapping, the genomic coordinates of genes (start and end positions) and SNPs were retrieved and used to define candidate sc-eQTLs (Figure 3a).

To illustrate biologically meaningful results, we note that the identified eQTLs in GMPs include both previously reported regulatory variants and novel candidate SNP–gene associations. For example, the SNP rs12116440 has been reported as a regulatory variant in peripheral blood mononuclear cells [23]. In contrast, associations such as RPS27-1:632647 and RPS27-1:28236165 represent novel cell-type-specific regulatory relationships uncovered by scQTLtools. In the case of RPS27-1:632647, increasing dosage of the alternative allele was associated with elevated gene expression (Figure 3c), whereas for RPS27-1:28236165, the alternative allele dosage showed a negative correlation with expression levels (Figure 3d).

Each identified eQTL is accompanied by model-specific statistical outputs. For the ZINB model, these include positions, eQTL-detected cell groups, fold change, and raw and adjusted *p*-values (Figure 3b). For the linear and Poisson models, summary statistics such as the regression coefficient, absolute effect size, *p*-value, and adjusted *p*-value are provided.

The results clearly reveal the cell-type-specific effects of eQTLs, demonstrating that certain regulatory effects emerge only in specific cellular contexts, such as in GMPs. These findings underscore the importance of cell-type-specific eQTL analysis and suggest potential mechanisms through which genetic variants may influence AML pathogenesis in a lineage-specific manner.

## 4. Discussion

In this study, we presented scQTLtools, a comprehensive R/Bioconductor package designed for identifying and visualizing sc-eQTLs from scRNA-seq data. scQTLtools supports a full analysis pipeline, from flexible data input and preprocessing to model-based sc-eQTLs detection and downstream visualization. By integrating standard single-cell data structures, supporting multiple genotype encoding formats, and providing built-in human and mouse genome annotation interfaces, scQTLtools streamlines sc-eQTL analysis and enhances accessibility for users across diverse research contexts.

Compared with existing tools such as Matrix eQTL, SCeQTL, eQTLsingle, and iBMQ, scQTLtools offers broader functionality, particularly in its ability to model zero-inflated expression patterns common in UMI-based scRNA-seq data and in its flexible support for multiple statistical frameworks. In addition, its preprocessing functions—covering gene/SNP filtering, expression normalization, and cell-level grouping—ensure robust and biologically meaningful results. The integrated visualization functions, with options for stratified plots across genotype groups and cell types, facilitate the interpretation of eQTL signals in heterogeneous cellular populations.

We demonstrated the utility of scQTLtools in a case study using matched scRNA-seq and genotype data from AML samples. Our analysis identified cell-type-specific eQTLs in GMPs, some of which showed opposing regulatory directions across different loci. These results highlight the power of scQTLtools to uncover cell-type-specific regulatory variation that would be missed by bulk eQTL analyses. The ability to dissect such variation may provide new insights into disease mechanisms and enable the identification of candidate functional variants.

Despite its strengths, scQTLtools has several limitations. First, it currently focuses on cis-eQTL mapping and does not support trans-eQTL detection, which may miss long-range regulatory effects. Second, although scQTLtools supports multiple statistical models, it does not yet implement covariate adjustment (e.g., batch effects or donor ID) within the eQTL testing framework. Nonetheless, the eQTLs identified by scQTLtools can serve as a valuable resource for downstream analyses. For example, integrating cell-type-specific eQTL results with genome-wide association study (GWAS) summary statistics may help prioritize putative causal variants and refine disease-associated loci. In addition, functional enrichment analysis of eQTL target genes can reveal dysregulated pathways in specific cell populations, providing insight into the biological processes underlying complex traits and diseases.

## 5. Conclusions

In conclusion, scQTLtools provides a robust, flexible, and user-friendly solution for sc-eQTL analysis, enabling researchers to investigate how genetic variation shapes gene expression at the single-cell level. By supporting multiple input formats and statistical models, the package facilitates scalable and biologically meaningful exploration of genotype–expression associations in complex tissues. One current limitation is that scQTLtools focuses exclusively on cis-eQTL detection. In future versions, we aim to incorporate trans-eQTL mapping to allow the detection of long-range regulatory effects. We anticipate that scQTLtools will serve as a valuable resource for the single-cell genomics community and facilitate functional interpretation of genetic variants in complex tissues and diseases.

## Figures and Tables

**Figure 1 biology-14-00743-f001:**
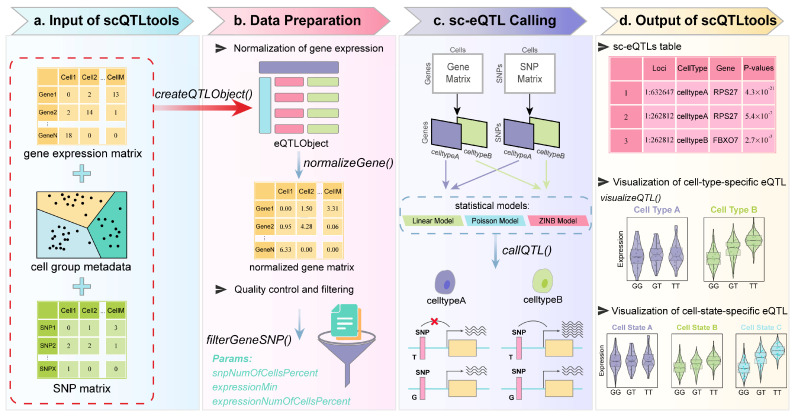
Overview of functionality and workflows with scQTLtools. (**a**) The input of scQTLtools. (**b**) The data preparation of scQTLtools. (**c**) The sc-eQTL calling of scQTLtools. (**d**) The output of scQTLtools (genotype groups: GG = homozygous reference, GT = heterozygous, TT = homozygous alternative).

**Figure 2 biology-14-00743-f002:**
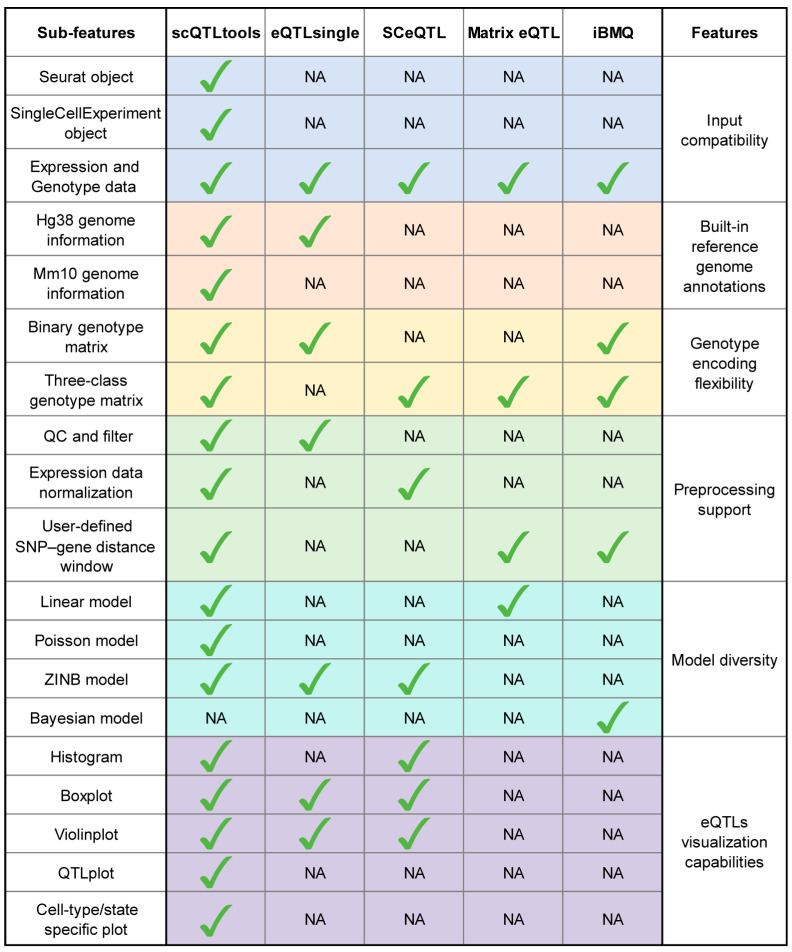
Comparison of supported features from currently available sc-eQTLs identification toolkits. “NA” indicates that the feature is not applicable to the tool; “Check mark” indicates that the feature is supported by the tool.

**Figure 3 biology-14-00743-f003:**
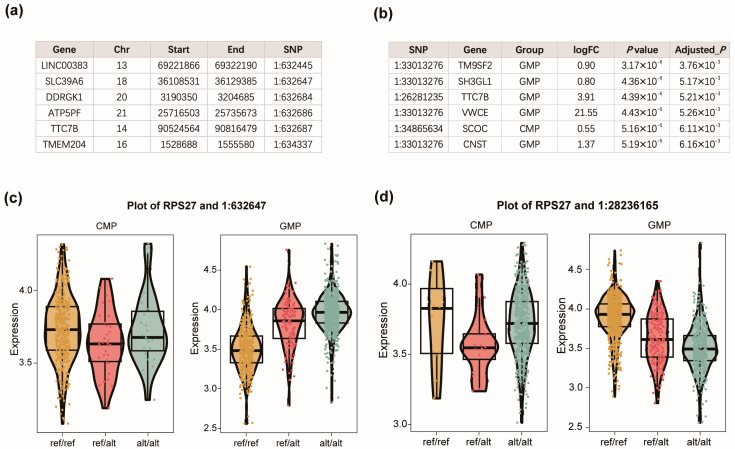
Case study demonstrating scQTLtools functionalities in sc-eQTL analysis. (**a**) Genomic coordinates of genes and SNPs used to detect candidate sc-eQTLs. (**b**) sc-eQTL results generated from scQTLtools. (**c**) Example of a cell-type-specific eQTL identified in GMP cells, where the alternative allele is associated with increased gene expression. GMP (granulocyte-monocyte progenitor) and CMP (common myeloid progenitor) are two major myeloid lineages analyzed in this study. (**d**) Another example in GMPs, where the alternative allele is associated with decreased gene expression.

## Data Availability

The AML scRNA-seq data are available in the Gene Expression Omnibus under accession GSE279118 (https://www.ncbi.nlm.nih.gov/geo/ (accessed on 30 May 2025)). The scQTLtools package can be downloaded from its Bioconductor page https://bioconductor.org/packages/scQTLtools/ (accessed on 30 May 2025). or the GitHub development page https://github.com/XFWuCN/scQTLtools (accessed on 30 May 2025).

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
