# Peer review of "scQTLtools: An R/Bioconductor Package for Comprehensive Identification and Visualization of Single-Cell eQTLs"

_biology, 2025, doi:10.3390/biology14070743_

Round 1
Reviewer 1 Report
Comments and Suggestions for Authors
This manuscript presents scQTLtools, a novel and comprehensive R/Bioconductor package designed for single-cell expression quantitative trait loci (sc-eQTL) analysis. The toolkit integrates preprocessing, modeling, and visualization functions within a user-friendly interface and supports multiple genotype encodings and statistical frameworks. It is a timely and well-executed contribution that addresses critical limitations in current tools, especially with regard to sparse data and flexible input support.
The software appears robust and is clearly documented. The manuscript is generally well-written and appropriately structured. A real-world application to acute myeloid leukemia (AML) data adds practical value and demonstrates the tool’s biological relevance.
Nonetheless, a few issues related to clarity, reproducibility, and performance benchmarking should be addressed before publication.
- The tool offers a rich feature set, but the manuscript could better emphasize what specifically distinguishes scQTLtools from closely related tools such as SCeQTL and eQTLsingle in practical usage scenarios.
- Include performance metrics (e.g., runtime, memory usage) or sensitivity analyses, especially under different sparsity conditions or model choices (linear, Poisson, ZINB).
- Briefly describe the sample characteristics and cell types included in the AML case study dataset to contextualize the analysis.
- Provide guidelines or diagnostic plots to help users choose between linear, Poisson, or ZINB models based on their data characteristics.
- Acknowledge the lack of trans-eQTL detection as a limitation more prominently in the abstract and conclusion.
- Briefly explain how scQTLtools handles missing genotype or expression values, especially in sparse matrices.
- Confirm if the package supports input from 10x Genomics-derived data and if preprocessing pipelines (e.g., CellRanger) are compatible.
- Discuss potential downstream applications of the results, e.g., integrating with GWAS summary statistics or functional enrichment tools.
- Line 111: Clarify “0/1/2/3 encoding” by explicitly stating what each numeric value corresponds to, even though it's explained later.
- Line 281–282: Slight rewording is needed. Replace “handle sparse scRNA-seq data using the ZINB model” with “model zero-inflated expression patterns common in UMI-based scRNA-seq data.”
Author Response
|
Comments 1: The tool offers a rich feature set, but the manuscript could better emphasize what specifically distinguishes scQTLtools from closely related tools such as SCeQTL and eQTLsingle in practical usage scenarios. Response 1: Thank you for this suggestion. We have revised the manuscript to better highlight the practical advantages of scQTLtools over SCeQTL and eQTLsingle. Specifically, we emphasize that scQTLtools supports multiple genotype encodings (binary and three-class), and provides integrated preprocessing and visualization modules that enhance usability and interpretability (Page 6, Lines 250-258). |
|
Comments 2: Include performance metrics (e.g., runtime, memory usage) or sensitivity analyses, especially under different sparsity conditions or model choices. Response 2: We appreciate this constructive suggestion. We have added a supplementary table (Table S1) reporting the runtime and peak memory usage of scQTLtools under three different modeling choices (linear, Poisson, and ZINB) using the AML dataset. To further evaluate robustness under different data characteristics, we conducted a sensitivity analysis by artificially downsampling the gene expression matrix to generate varying sparsity levels and assessed corresponding changes in performance metrics. These results are now summarized in the Results section and provide practical insights for users selecting models under different data sparsity conditions (Page 7, Lines 259-266). |
|
Comments 3: Briefly describe the sample characteristics and cell types included in the AML case study dataset to contextualize the analysis. Response 3: We thank the reviewer for this recommendation. We have added a paragraph in the Case Study section detailing the sample characteristics (Page 7, Lines 278-281). The AML dataset consists of three donor samples with matched genotype and scRNA-seq profiles, encompassing major myeloid lineages such as HSCs, GMPs, CMPs, and monocytes. This context helps clarify the cell-type-specific findings described later. |
|
Comments 4: Provide guidelines or diagnostic plots to help users choose between linear, Poisson, or ZINB models based on their data characteristics. Response 4: We appreciate the reviewer’s insightful suggestion. In response, we have added brief guidelines in the Materials and Methods section to help users select an appropriate model based on gene-level expression characteristics. Specifically, we describe when the linear, Poisson, or ZINB models are most suitable depending on data sparsity and distribution (Page 5, Lines 187-192). |
|
Comments 5: Acknowledge the lack of trans-eQTL detection as a limitation more prominently in the abstract and conclusion. Response 5: We agree with this point and have revised both the Abstract and Conclusions sections to explicitly state that the current version of scQTLtools is limited to cis-eQTL detection. We have also noted that future development will aim to incorporate trans-eQTL mapping (Page 1, Lines 38-39; Page 9, Lines 352-356). |
|
Comments 6: Briefly explain how scQTLtools handles missing genotype or expression values, especially in sparse matrices. Response 6: We thank the reviewer for this important question. To address missing genotype or expression values, scQTLtools provides filtering functions during preprocessing that allow users to exclude low-information SNPs and genes based on customizable thresholds. This helps minimize the impact of missing values on downstream modeling. Additionally, for sparse scRNA-seq data characterized by high proportions of zero values, scQTLtools implements the zero-inflated negative binomial (ZINB) model, which has been widely reported as effective for modeling sparse, zero-inflated single-cell expression data. These design choices ensure robust performance when analyzing incomplete or sparse datasets. |
|
Comments 7: Confirm if the package supports input from 10x Genomics-derived data and if preprocessing pipelines (e.g., CellRanger) are compatible. Response 7: We confirm that scQTLtools is compatible with 10x Genomics-derived data. Users can directly import gene expression matrices generated by CellRanger using Seurat or SingleCellExperiment objects, which are fully supported by our input functions. We have clarified this in the Data preparation and input functions section of the manuscript (Page 3, Lines 112-114). |
|
Comments 8: Discuss potential downstream applications of the results, e.g., integrating with GWAS summary statistics or functional enrichment tools. Response 8: We have expanded the Discussion section to highlight potential downstream applications (Page 10, Lines 342-348). These include integration of identified eQTLs with GWAS summary statistics to identify putative causal variants, and functional enrichment analyses of eQTL target genes to uncover pathway-level insights. |
|
Comments 9: Line 111: Clarify “0/1/2/3 encoding” by explicitly stating what each numeric value corresponds to. Response 9: We have revised the corresponding description in the Data Preparation and Input Functions section to clarify that: 1 indicates a homozygous reference genotype, 2 indicates a homozygous alternative genotype, 3 indicates a heterozygous genotype, and 0 indicates missing values (Page 3, Lines 116-119). |
|
Comments 10: Line 281–282: Slight rewording is needed. Replace “handle sparse scRNA-seq data using the ZINB model” with “model zero-inflated expression patterns common in UMI-based scRNA-seq data.” Response 10: Thank you for the wording suggestion. We have revised the sentence as recommended to improve precision and clarity (Page 9, Lines 324-325). |

Reviewer 2 Report
Comments and Suggestions for Authors
A brief summary
Undoubtedly, the study of genetic loci controlling gene expression, known as eQTL (expression quantitative trait loci), is a very important tool alongside GWAS. The interpretation of GWAS can be complicated when variants are located in genes that do not cause disease themselves but regulate the activity of driver genes. In such cases, identifying eQTL becomes a means of accurately pinpointing regulatory relationships. The comparison of the tool developed by the authors, scQTLtools, with existing tools (eQTLsingle, SCeQTL, Matrix eQTL, iBMQ) and the expansion of its functionality is of particular value, as it will enhance the quality of analysis of single-cell gene expression and SNP data, as well as the level of research in which this tool will be employed. Among the 20 literature sources, 10 are from the last 5 years. No issues of self-citation were identified.
General concept comments
- The authors mention the difference between bulk and single-cell analysis. It is necessary to add one or two sentences to emphasize that the joint analysis of single-cell gene expression and SNP data is possible due to modern technologies, allowing researchers to obtain biological data from different modalities from the same cell. This should highlight the difference between data from two different single cells and from the same one.
- The authors mention the use of the S4 format. While S4 objects are commonly used to store and manage complex single-cell data, considering the aim to improve and expand the functional characteristics of the new tool, it would be worthwhile to consider the capability of working with the h5ad (HDF5) format. This would be beneficial in cases where the user might want to conduct some analysis based on Python.
- It is clear that the paper is dedicated to bioinformatics. However, the journal's title suggests that equal attention should be given to the biological aspects of the issue. For instance, the dataset used to test the tool focuses on human acute myeloid leukemia (AML) – a serious disease. It would be very important for readers to know what was previously known about the identified SNPs before using scQTLtools. Are these fundamentally new insights or clarifications of previous findings?
- Is the code for the functions createQTLObject() different from the analogous function in the Seurat package – CreateSeuratObject(), or did the authors write their own function? The same question applies to normalizeGene()/NormalizeData().
- The number of citations for the tools mentioned in the article is Matrix eQTL – 1800 (2012), iBMQ – 13 (2016), SCeQTL – 12 (2020), eQTLsingle – 16 (2021), which indicates that interest in this approach has declined since its inception. This may be linked to insufficient justification for the results obtained. All works describe the tool itself and its potential applications in research. However, most users face challenges in choosing parameters at each stage of the analysis. A section with adapted information about the mathematical transformations used would significantly enhance the appeal of the publication.
Specific comments
- In Figure 1, the legend lacks an explanation for GG, GT, and TT.
- In Figure 3, the legend lacks an explanation for GMP and CMP. Of course, the explanation can be found in the text, but only GMP is defined in the text. There is a difficulty in distinguishing individual points against the background of the violin plots presented. Since the points are an important part of the visualization, changes need to be made to improve their readability.
Author Response
|
General Comment 1: The authors mention the difference between bulk and single-cell analysis. It is necessary to add one or two sentences to emphasize that the joint analysis of single-cell gene expression and SNP data is possible due to modern technologies, allowing researchers to obtain biological data from different modalities from the same cell. This should highlight the difference between data from two different single cells and from the same one. Response 1: We thank the reviewer for this insightful suggestion. We have revised the Introduction to highlight that modern single-cell multi-omics technologies (Page 2, Lines 54-59), such as joint profiling of gene expression and chromatin accessibility, enable simultaneous measurement of different modalities within the same cell. This integrated approach avoids ambiguity caused by separately acquired data from different cells, thereby improving the resolution and accuracy of eQTL analyses. |
|
General Comment 2: The authors mention the use of the S4 format. While S4 objects are commonly used to store and manage complex single-cell data, considering the aim to improve and expand the functional characteristics of the new tool, it would be worthwhile to consider the capability of working with the h5ad (HDF5) format. Response 2: We appreciate this valuable comment. scQTLtools is currently designed for the R/Bioconductor ecosystem and adopts the S4 class structure to maintain compatibility with widely used R-based single-cell workflows. However, we recognize the increasing popularity of Python-based workflows and formats such as h5ad. We plan to incorporate h5ad support in a future version of scQTLtools to facilitate interoperability with Python-based tools such as Scanpy. |
|
General Comment 3: It would be very important for readers to know what was previously known about the identified SNPs before using scQTLtools. Are these fundamentally new insights or clarifications of previous findings? Response 3: Thank you for this suggestion. We have revised the Case Study section to clarify that the identified eQTLs in GMPs include both previously reported regulatory variants and novel candidate SNP-gene associations (Page 8, Lines 291-294). Where applicable, we provide references to prior studies. This contextualization underscores the potential of scQTLtools to validate known regulatory loci and uncover new ones in specific cell populations. |
|
General Comment 4: Is the code for the functions createQTLObject() different from the analogous function in the Seurat package – CreateSeuratObject(), or did the authors write their own function? The same question applies to normalizeGene()/NormalizeData(). Response 4: Thank you for raising this question. We confirm that the functions createQTLObject() and normalizeGene() in scQTLtools were independently developed by our team and are not derived from Seurat’s functions. We intentionally adopted similar naming conventions to improve usability for users familiar with single-cell workflows. The function createQTLObject() serves as a data entry point that integrates gene expression, genotype, and metadata for downstream single-cell eQTL analysis. This design is conceptually similar to what CreateSeuratObject() does for general scRNA-seq workflows, but the underlying implementation and analytical purpose are specifically tailored for eQTL modeling. Similarly, the normalizeGene() function supports a broader set of normalization methods, including logNormalize, CPM, TPM, DESeq, and limma, and is optimized for genotype-expression association analysis, whereas Seurat’s NormalizeData() provides more limited options mainly for general single-cell expression processing. |
|
General Comment 5: The number of citations for the tools mentioned in the article is Matrix eQTL – 1800 (2012), iBMQ – 13 (2016), SCeQTL – 12 (2020), eQTLsingle – 16 (2021), which indicates that interest in this approach has declined since its inception. This may be linked to insufficient justification for the results obtained. All works describe the tool itself and its potential applications in research. However, most users face challenges in choosing parameters at each stage of the analysis. A section with adapted information about the mathematical transformations used would significantly enhance the appeal of the publication. Response 5: We thank the reviewer for this constructive comment. To address the concern about usability and model interpretability, we have expanded the “sc-eQTLs mapping and statistical models” section to include practical guidelines for selecting among the three supported statistical models (Page 5, Lines 187-192). In addition, we plan to include a new diagnostic function, plotModelDiagnostics(), in a future release of scQTLtools to help users visualize gene-level expression distributions and assess sparsity, thereby facilitating informed model selection. Regarding the citation numbers of existing tools, we agree that Matrix eQTL has remained widely used (over 1800 citations) despite not being designed for single-cell data, whereas SCeQTL and eQTLsingle have received comparatively few citations. Our survey of recent sc-eQTL studies suggests that many researchers still rely on Matrix eQTL or custom scripts to perform single-cell eQTL analyses. This trend may reflect a lack of fully featured and user-friendly tools rather than a decline in interest in sc-eQTL methods perse. Notably, both SCeQTL and eQTLsingle have certain functional limitations, such as restricted genotype encoding, lack of preprocessing modules, or insufficient visualization support. scQTLtools was developed specifically to address these gaps and provide a more comprehensive and accessible platform for sc-eQTL analysis. |
|
Specific Comment 1: In Figure 1, the legend lacks an explanation for GG, GT, and TT. Response 6: Thank you for pointing this out. We have updated the legend of Figure 1 to explicitly define the genotype groups GG (homozygous reference), GT (heterozygous), and TT (homozygous alternative). (Page 4, Lines 132-133) |
|
Specific Comment 2: In Figure 3, the legend lacks an explanation for GMP and CMP. Of course, the explanation can be found in the text, but only GMP is defined in the text. There is a difficulty in distinguishing individual points against the background of the violin plots presented. Since the points are an important part of the visualization, changes need to be made to improve their readability. Response 7: We have updated the legend of Figure 3 to clearly define GMP (granulocyte-monocyte progenitor) and CMP (common myeloid progenitor). Additionally, we have improved the readability of the violin plots by adjusting point transparency and size to enhance visibility of individual data points (Page 9, Lines 312-313). |
